# Are Advanced Glycation End Products in Skin Associated with Vascular Dysfunction Markers? A Meta-Analysis

**DOI:** 10.3390/ijerph17186936

**Published:** 2020-09-22

**Authors:** Alicia Saz-Lara, Celia Álvarez-Bueno, Vicente Martínez-Vizcaíno, Blanca Notario-Pacheco, Irene Sequí-Dominguez, Iván Cavero-Redondo

**Affiliations:** 1Health and Social Research Center, Universidad de Castilla-La Mancha, 16171 Cuenca, Spain; Alicia.delSaz@uclm.es (A.S.-L.); vicente.martinez@uclm.es (V.M.-V.); Blanca.Notario@uclm.es (B.N.-P.); irene.sequidominguez@uclm.es (I.S.-D.); ivan.cavero@uclm.es (I.C.-R.); 2Universidad Politécnica y Artística del Paraguay, 001518 Asuncion, Paraguay; 3Facultad de Ciencias de la Salud, Universidad Autónoma de Chile, 3460000 Talca, Chile

**Keywords:** cardiovascular disease, skin autofluorescence, advanced glycation end products, arterial stiffness, pulse wave velocity, carotid intima media thickness

## Abstract

Evidence exists regarding the association between advanced glycation end products and different cardiovascular disease subclinical processes, such as arterial stiffness and atherosclerosis. With this systematic review and meta-analysis, we aimed to provide a synthesis of the evidence regarding the association of arterial stiffness measured by pulse wave velocity and atherosclerosis measured by carotid intima media thickness with skin autofluorescence. A systematic search was performed using: MEDLINE (PubMed), SCOPUS, and Web of Science, until 30 March 2020. Cross-sectional studies or baseline data from prospective longitudinal studies were considered. The DerSimonian and Laird method was used to calculate the pooled estimates of correlation coefficients and the corresponding 95% confidence intervals (CI) for the association of pulse wave velocity and carotid intima media thickness with skin autofluorescence. Twenty-five studies were included in the systematic review and meta-analysis, including 6306 subjects. The pooled correlation coefficient was 0.25 (95% CI: 0.18, 0.31) for pulse wave velocity and skin autofluorescence, and 0.31 (95% CI: 0.25, 0.38) for carotid intima media thickness and skin autofluorescence. This systematic review and meta-analysis provide a synthesis of the evidence showing a positive weak association of pulse wave velocity and carotid intima media thickness with skin autofluorescence.

## 1. Introduction

Advanced glycation end products (AGEs) are a group of molecules that, through non-enzymatic glycation reactions and stimulation of oxidative stress [1,2], are involved in the development of cardiovascular diseases (CVDs) [3]. Evidence suggests that the accumulation of AGEs in tissues, measured by skin autofluorescence (SAF) [4], increases with age and smoking [5], as well as in individuals with high levels of inflammation or diabetic conditions [6]. AGEs are involved in the progression of atherosclerosis and some chronic diseases, such as chronic renal failure, Alzheimer’s disease, and diabetes mellitus [2,5,7]. Finally, AGEs have also been associated with endothelial dysfunction and early vascular aging [6].

The aged vascular system is characterized by decreased elasticity of vessels, accumulation of interstitial collagen, increased arterial stiffness, and thickening of the carotid intima media [8], amongst other factors. Traditionally, both arterial stiffness and atherosclerosis have been considered good predictors of increased cardiovascular risk [9]. Arterial stiffness is one of the first changes detected in the wall of arteries, in both function and structure, playing an important role in CVD [10].

Evidence exists of the association between AGE levels and different CVD subclinical processes, such as arterial stiffness and atherosclerosis [11,12]. The accumulation of AGEs has been associated with an increase in pulse wave velocity (PWv), which is the gold standard for measuring arterial stiffness [13], both in healthy adults and individuals with different pathologies, such as patients with coronary heart disease, type 1 or 2 diabetes mellitus, and kidney disease [14,15,16,17,18,19]. Likewise, a positive correlation between atherosclerosis measured using carotid intima media thickness (C-IMT) and AGEs in both non-diabetic and diabetic patients has been reported [20,21,22]. Even though C-IMT is no longer included as a factor of CVD risk according to international guidelines [23], its assessment allows for knowledge of arterial thickening in preclinical phases of disease, before the vessel lumen becomes compromised [24]. However, the evidence for the association between AGEs and these subclinical markers of atherosclerosis and arterial stiffness is currently sparse and inconclusive.

Therefore, the purpose of this systematic review and meta-analysis was to provide a synthesis of the evidence on the association of arterial stiffness (measured using PWv) and atherosclerosis (measured through C-IMT) with AGE levels measured by SAF, which has been proposed as a novel and effective indicator to help with the prevention of CVD. This knowledge will contribute important evidence for assessing the cardiovascular risk derived from these subclinical processes.

## 2. Materials and Methods 

This systematic review and meta-analysis are reported according to the Meta-analysis of Observational Studies in Epidemiology statement (MOOSE) [25] and the Preferred Reporting Items for Systematic Reviews and Meta-analyses (PRISMA) [26], and performed following the Cochrane Collaboration Handbook recommendations [27]. This study was registered in the International Prospective Register of Systematic Reviews (registration number: CRD42020168219).

### 2.1. Search Strategy

A systematic search of studies was conducted through three databases: MEDLINE (via PubMed), Scopus, and Web of Science, from their inception to 30 March 2020 (Appendix A). The following keywords were used to perform the search: “Advanced glycation end products”, “AGEs”, “pulse wave velocity”, “PWv”, “arterial stiffness”, “intima media thickness”, “IMT”, “vascular stiffness”, “arterial health”, “vascular function”, “endothelial function”, “cardiovascular disease”, “cardiovascular risk”, “skin autofluorescence”, and “SAF”. The list of references of the included articles as well as of previous systematic reviews or meta-analyses were searched. A last search was conducted just before the final analysis to include the most recently published studies.

### 2.2. Study Selection

The included studies assessed the association of PWv and C-IMT with SAF. Inclusion criteria were (1) studies including individuals over 18 years of age, (2) AGEs measured using SAF, (3) arterial stiffness measured using PWv, (4) C-IMT measured using ultrasound techniques, and (5) cross-sectional studies or baseline data of prospective longitudinal studies. Studies were excluded if (1) outcome measures were not reported as correlation values for the association between SAF and PWv or C-IMT; (2) they were review articles, editorials, or case reports; or (3) they were not written in English or Spanish.

The studies were selected independently by two researchers (A.S.-L., I.C.-R.). All abstracts of the retrieved studies were examined and studies that did not meet the previously described eligibility criteria were excluded. Disagreements between researchers were solved by consensus or by a third researcher (C.Á.-B.).

### 2.3. Search and Data Extraction

The main characteristics of the included studies are summarized in Table 1, which includes information on (1) the first author and year of publication; (2) country where study data were collected; (3) study design; (4) sample characteristics (sample size, average age, population type, and body mass index (BMI)); (5) outcome: PWv (type of PWv (aortic PWv, brachial ankle PWv, carotid femoral PWv, carotid radial PWv), measuring device, PWv mean values) or C-IMT (measuring device, C-IMT mean values); and (6) SAF (measuring device, SAF mean).

### 2.4. Quality Assessment and Potential Bias

Methodological quality was assessed using the Quality Assessment Tool for Observational Cohort and Cross-Sectional Studies from the United States National Institute of Health National Heart, Lung, and Blood Institute [28]. This tool evaluates the risk of bias according to the following domains: Quality of the research question, reporting of the population definition, participation rate, recruitment, sample size, appropriateness of statistical analyses, timeframe for associations, exposure levels, ascertainment of the exposure, appropriateness of the outcome measured, outcome blinding of researchers, loss to follow-up, and confounding variables. The general bias of each study was considered “good” if most criteria were met and with a low risk of bias; “fair” if some criteria were met and with a moderate risk of bias; or “poor” if few criteria were met and with a high risk of bias.

Once the information regarding the authors, date, and sources of each included manuscripts were blinded, two researchers (A.S.-L., I.C.-R.) independently extracted data and assessed quality. Disagreements were solved by consensus or through the intervention of a third researcher (C.Á.-B.).

### 2.5. Statistical Analysis

Forest plots are used to show correlation estimates (Pearson and Spearman coefficients) for the association between PWv with SAF, and C-IMT with SAF. The DerSimonian and Laird method was used to calculate the pooled estimates of correlation coefficients and their respective 95% confidence intervals (CIs) [29]. Heterogeneity was examined using the I2 statistic, which ranges between 0% and 100% [30]. According to the I2 values, heterogeneity was considered not important (0% to 40%), moderate (30% to 60%), substantial (50% to 90%), or considerable (75% to 100%). The corresponding *p*-values were also considered.

Sensitivity analyses (systematic re-analyses while removing each study one at a time) were conducted to assess the robustness of the summary estimates. A sensitivity analyses between each of the outcomes (PWv and C-IMT) with SAF were conducted considering diabetic patients only. Additionally, a sensitivity analysis was performed including only studies with a control group. Random effects meta-regression analyses were performed to determine whether age, sex, BMI, total cholesterol, high-density lipoprotein (HDL), low-density lipoprotein (LDL), triglycerides, systolic blood pressure (SBP), diastolic blood pressure (DBP), or glycated hemoglobin A1c (HbA1c) are significant moderators of the association between PWv and SAF, and C-IMT and SAF.

Finally, publication bias was assessed through Egger’s regression asymmetry test [31]. A *p*-value of <0.10 was used to determine if there was significant publication bias.

Analyses were performed with Stata 15.0 (Stata, College Station, TX, USA).

## 3. Results

### 3.1. Systematic Review

The flowchart of this systematic review and meta-analysis is presented in Figure 1. The search retrieved a total of 90 articles, of which 25 studies [12,13,15,16,17,18,20,21,22,32,33,34,35,36,37,38,39,40,41,42,43,44,45,46,47] were selected for inclusion in the systematic review according to the inclusion criteria. Finally, 12 studies evaluating the association between PWv and SAF and 17 studies evaluating the association between C-IMT and SAF were identified and considered in the meta-analysis. From the studies included in the systematic review, five presented results for both PWv and C-IMT.

The descriptive characteristics of the included studies are shown in Table 1 and Appendix A. Studies were published between 2006 and 2018, were cross-sectional (22 studies) or prospective longitudinal (three studies) [41,42,44], with sample sizes ranging from 38 to 1717 subjects (aged 18 to 80 years), and were performed in Europe (14 studies) and Asia (11 studies). All studies included both men and women, except for two including men only and one including women only. Regarding the type of population, nine studies included participants with diabetes mellitus (type 1 or 2), four included individuals with different stages of chronic kidney disease, and the remaining studies included individuals with different types of pathologies, such as vascular, metabolic, or autoimmune pathologies. 

**Table 1 ijerph-17-06936-t001:** Characteristics of the studies included in the systematic review and meta-analysis of the relationship between pulse wave velocity or carotid intima media thickness and skin autofluorescence.

Study (Year)	Country	Study Design	Population Characteristics				Outcome			Skin Autofluorescence	
			Sample Size (*n*)	Age (Years)	Type of Population	BMI (kg /m^2^)	Type	Measuring Device	Mean (mm)	Measuring Device	Mean (AU)
Araszkiewicz et al., 2015 [42]	Poland	Prospective longitudinal	UHS: 77	UHS: 23 (20–28)	DM type 1	UHS: 23 (21–25)	C-IMT	Acuson Cv 70	UHS: 0.57 (0.52–0.67)	AGE Reader	UHS: 2.2 (1.9–2.6)
Hollander et al., 2007 [34]	The Netherlands	Cross-sectional	UHS: 8 HS: 30	UHS: 27 (20–34) HS: 25 (21–32)	Glycogen storage disease type Ia	UHS: 24 (22–28) HS: 23 (21–26)	C-IMT	Acuson 128 XP	UHS: 0.53 (0.48–0.59) HS: 0.6 (0.58–0.62)	AGE Reader	UHS: 1.67 (1.57–1.76) HS: 1.55 (1.30–1.76)
Blaauw et al., 2006 [32]	The Netherlands	Cross-sectional	UHS: 26 HS: 17	UHS: 30.0 ± 4.0 HS: 32.0 ± 3.0	Preeclampsia	UHS: 25.0 ± 5.0 HS: 23.0 ± 3.0	C-IMT	Acuson 128 XP	UHS: 0.64 ± 0.07 HS: 0.63 ± 0.09	USB2000	NA
Llaurado et al., 2014 [16]	Spain	Cross-sectional	UHS: 68 HS: 68	UHS: 35.3 ± 10.1 HS: 35.4 ± 10.2	DM type 1	UHS: 25.7 ± 3.6 HS: 24.0 ± 3.1	a-PWV	Millar tonometer: SPC-301	UHS: 6.8 (6.0–7.9) HS: 6.1 (5.5–6.7)	AGE Reader	UHS: 2.1 (1.8–2.3) HS: 1.7 1.6–2.1)
Osawa et al., 2017 [21]	Japan	Cross-sectional	UHS: 105 HS: 23	UHS: 37.4 ± 12.4 HS: 34.7 ± 6.2	DM type 1	UHS: 23. 0 ± 3.0 HS: 20.6 ± 2.6	ba-PWV	BP203RPE	UHS: 13.18 ± 2.48 HS: 12.25 ± 1.56	AGE Reader	UHS: 2.07 ± 0.50 HS: 1.90 ± 0.26
							C-IMT	NA	UHS: 1.09 ± 0.48 HS: 0.76 ± 0.21		
De Leeuw et al., 2007 [33]	The Netherlands	Cross-sectional	UHS: 55 HS: 55	UHS: 43.0 ± 12. HS: 43.0 ± 13.0	Systemic lupus erythematosus	UHS: 24.3 ± 4.0 HS: 24.7 ± 4.0	C-IMT	NA	UHS: 0.67 ± 0.16 HS: 0.69 ± 0.15	AGE Reader	UHS: 1.50 ± 0.5 HS: 1.28 ± 0.4
De Groot et al., 2015 [44]	The Netherlands	Prospective longitudinal	UHS: 58 HS: 58	UHS: 18–80 HS: 18–80	Rheumatoid arthritis	NA	C-IMT	NA	UHS: 0.73 (0.45–1.64) HS: 0.72 (0.39–1.46)	AGE Reader	UHS: 2.55 (1.29–4.65) HS: 2.12 (1.32–3.82)
De Leeuw et al., 2010 [36]	The Netherlands	Cross-sectional	UHS: 24 HS: 21	UHS: 51.0 ± 11.0 HS: 56.0 ± 14.0	Systemic autoimmune Disease (Wegener’s granulomatosis)	UHS: 25.0 ± 3.0 HS: 26.0 ± 5.0	C-IMT	NA	UHS: 0.72 (0.62–0.81) HS: 0.67 (0.59–0.79)	NA	UHS: 1.5 ± 0.5 HS: 1.3 ± 0.3
Den Dekker et al., 2013 [39]	The Netherlands	Cross-sectional	UHS1: 67 UHS2: 60 HS: 96	UHS1: 51.8 ± 7.8 UHS2: 63.5 ± 7.6 HS: 43.8 ± 9.5	Atherosclerosis	UHS1: 26.6 (23.8–29.8) UHS2: 26.3 (24.2–29.5) HS: 25.0 (23.1–27.7)	C-IMT	Acuson 128 XP	UHS1: 0.83 (0.67–0.98) UHS2: NA HS: 0.65 (0.57–0.74)	AGE Reader	UHS1: 2.11 (1.83–2.46) UHS2: 2.71 (2.15–3.27) HS: 1.87 (1.68–2.12)
Dadoniene et al., 2015 [43]	Lithuania	Cross-sectional	UHS: 47 HS: 47	UHS: 52.64 ± 11.2 HS: 52.57 ± 7.69	Systemic sclerosis	UHS: 24.27 ± 4.63 HS: 26.09 ± 4.50	cr-PWV	SphygmoCor	UHS: 7.53 ± 1.70 HS: 7.51 ± 1.30	AGE Reader	UHS: 2.23 ± 0.54 HS: 1.90 ± 0.47
Lutgers et al., 2010 [20]	The Netherlands	Cross-sectional	UHS: 59	UHS: 55.0 ± 10.0	Healthy Metabolic syndrome	UHS: 24.9 ± 2.5	C-IMT	Acuson 128 XP	UHS: 0.8 ± 0.15	NA	UHS: 1.57 ± 0.41
Ueno et al., 2008 [17]	Japan	Cross-sectional	UHS: 120 HS: 110	UHS: 58.1 ± 9.3 HS: 57.0 ± 10.5	End-stage renal disease	NA	ba-PWV	BP203RPE	UHS: 17.92 ± 4.49 HS: 14.21 ± 2.26	AGE Reader	UHS: 1.8 ± 0.7 HS: 1.3 ± 0.5
Ninomiya et al., 2018 [13]	Japan	Cross-sectional	UHS: 140	UHS: 59.3 ± 12.8	DM type 1 and type 2	UHS: 27.3 ± 5.5	ba-PWV	BP203RPE	UHS: 16.98 ± 4.04	AGE Reader	UHS: 2.5 ± 0.5
							C-IMT	NA	SS: 1.8 ± 0.8		
Ueno et al., 2011 [38]	Japan	Cross-sectional	UHS: 212	UHS: 59.9 ± 10.1	End-stage renal disease	UHS: 21.4 ± 2.8	NA	NA	NA	AGE Reader	UHS: 1.8 ± 0.7
							C-IMT	NA	UHS: 0.762 ± 0.163		
Hangai et al., 2016 [45]	Japan	Cross-sectional	UHS: 122	UHS: 61.0 ± 13.0	DM type 2	UHS: 26.4 ± 5.1	ba-PWV	BP203RPE	UHS: 15.69 ± 3.11	AGE Reader	UHS: 2.42 ± 0.417
							C-IMT	LOGIQ 500	UHS: 1.64 ± 0.70 mm		
Osawa et al., 2018 [12]	Japan	Cross-sectional	UHS: 193 HS: 24	UHS: 61.1 ± 12.3 HS: 40.3 ± 7.8	DM type 2	UHS: 27.7 ± 5.95 HS: 20.9 ± 2.9	ba-PWV	BP203RPE	UHS: 17.19 ± 4.58 HS: 12.75 ± 1.38	AGE Reader	UHS: 2.57 ± 0.47 HS: 1.91 ± 0.29
							C-IMT	NA	SS: 1.89 ± 0.78 HS: 0.92 ± 0.54		
Yoshioka, 2018 [47]	Japan	Cross-sectional	UHS: 162 HS: 42	UHS: 61.2 ± 11.2 HS: 53.8 ± 13.0	DM type 2	UHS: 24.9 ± 4.0 HS: 22.6 ± 4.0	C-IMT	NA	UHS: 1.64 ± 0.73 HS: 1.10 ± 0.23	AGE Reader	UHS: 2.53 ± 0.45 HS: 2.19 ± 0.34
Tanaka et al., 2009 [35]	Japan	Cross-sectional	UHS: 128 HS: 19	UHS: 65.1 ± 11.6 HS: 64.1 ± 12.4	Chronic kidney disease (DM)	UHS: 22.1 ± 3.3 HS: 24.6 ± 3.2	C-IMT	SSD- 5000	UHS: 0.9 ± 0.4 HS: NA	AGE reader	UHS: 2.35 ± 0.68 HS: 1.30 ± 0.37
Kimura et al., 2014 [41]	Japan	Prospective longitudinal	UHS: 86	UHS: 65.1 ± 11.6	DM, primary glomerulonephritis, Hypertension, and other diseases	UHS: 22.1 ± 3.3	C-IMT	SSD- 5000	UHS: 0.9 ± 0.4	AGE reader	UHS: 2.35 ± 0.68
Temma et al., 2015 [22]	Japan	Cross-sectional	UHS: 61	UHS: 66.6 ± 9.2	DM type 2	UHS: 25.5 ± 4.6	C-IMT	GM-72P00A	UHS: 1.64 ± 0.75 mm	AGE Reader	UHS: 2.50 ±0.50
Hofmann et al., 2013 [18]	Germany	Cross-sectional	UHS: 52	UHS: 68.7 ± 10.15	Coronary heart disease	UHS: 27.8 ± 4.0	a-PWV	Vicorder	NA	AGE Reader	NA
McIntyre et al., 2011 [37]	United Kingdom	Cross-sectional	UHS: 284 HS: 1423	UHS: 73.5 ± 8.0 HS: 72.8 ± 9.0	Chronic kidney disease stage 3	NA	cf-PWV	Vicorder	UHS: 10.4 ± 2.0 HS: 9.8 ± 2.0	AGE Reader	UHS: 3.0 ± 0.7 HS: 2.7 ± 0.6
McIntyre et al., 2013 [40]	Switzerland	Cross-sectional	UHS: 1717	UHS: 74 (67–79)	Chronic kidney disease	UHS: 28.4 (25.6–31.8)	a-PWV	Vicorder	UHS: 9.9 ± 2.0	AGE Reader	UHS: 2.67 (2.30–3.07)
Igase et al., 2017 [46]	Japan	Cross-sectional	UHS: 18 HS: 208	UHS: 76.5 ± 6.7 HS: 67.2 ± 9.9	Mild cognitive impairment	UHS: 22.9 ± 2.1 HS: 22.8 ± 3.0	ba-PWV	FORM/ABI	UHS: 17.7 ± 3.19 HS: 15.77 ± 2.77	AGE Reader	UHS: 2.56 ± 0.55 HS: 2.10 ± 0.41
Watfa et al., 2012 [15]	France	Cross-sectional	HS1: 55 HS2: 61	HS1: 49.1 ± 10.4 HS2: 77.5 ± 8.4	Healthy	HS1: 27.2 ± 5.5 HS2: 25.3 ± 4.5	cf-PWV	Pulse pen device (DiaTecne srl)	HS1: 7.48 ± 1.92 HS2: 11.83 ± 4.17	AGE Reader	HS1: 2.11 ± 0.45 HS2: 2.75 ± 0.6

Data are shown as mean ± SD or interquartile range: a-PWV: Aortic pulse wave velocity; AU: Arbitrary units; ba-PWV: Brachial ankle pulse wave velocity; BMI: Body max index; cf-PWV: Carotid femoral pulse wave velocity; C-IMT: Carotid intima media thickness; cr-PWV: Carotid radial pulse wave velocity; DM: Diabetes mellitus; HS: Healthy subjects; NA: Not available; PWV: Pulse wave velocity; SD: Standard deviation; UHS: Unhealthy subjects.

### 3.2. Quality Assessment and Potential Bias

The overall risk of bias of studies reporting the association between SAF and PWv was moderate in nearly two-thirds of the studies (61.5%). None of the studies provided information regarding the justification of the sample size or the blinding of evaluators. Additionally, all included studies were cross-sectional, affecting the domains related to the exposure levels and time frame to observe an effect. Finally, the domains evaluating repeated exposure and follow-up rate could not be evaluated (Appendix A).

The overall risk of bias of studies reporting the association between SAF and C-IMT was moderate in over two-thirds of the studies (70.6%). A low risk of bias was observed in 17.6% of included studies, whereas 11.8% had a high risk of bias. Regarding each domain, 70.6% of included studies did not provide justification for the necessary sample size. For domains concerning exposure levels, 82.4% of the studies presented a high risk of bias due to their cross-sectional design, and consequently did not provide repeated exposure assessments or follow-up rates. Additionally, 88.2% of the studies did not report whether the evaluator was blinded and the duration of the follow-up to see an effect (Appendix A).

### 3.3. Meta-Analysis

Figure 2 shows the correlation coefficients between PWv and SAF, and C-IMT and SAF. The pooled correlation coefficient estimate was 0.25 (95% CI: 0.18, 0.31) for PWv and SAF with substantial heterogeneity (I2: 76.3%; *p* < 0.001), and 0.31 (95% CI: 0.25, 0.38) for C-IMT and SAF with moderate heterogeneity (I2 = 49.9%; *p* = 0.007).

### 3.4. Sensitivity and Meta-Regression Analyses.

The pooled correlation coefficient estimate was not significantly different (in magnitude or direction) when individual study data were removed from the analyses one at a time.

Figure 3 shows the correlation coefficients between PWv and SAF, and C-IMT and SAF for the analyses performed considering diabetic patients only. The pooled correlation coefficient estimate for PWv and SAF was 0.31 (95% CI: 0.21, 0.42), with moderate heterogeneity (I2: 50.0%; *p* = 0.091). For C-IMT and SAF, the pooled correlation coefficient estimate was 0.29 (95%CI: 0.22, 0.37), with moderate heterogeneity (I2 = 40.0%; *p* = 0.112). 

Appendix A shows the correlation coefficients between PWv and SAF, and C-IMT and SAF for the analyses that included only studies with a control group. The pooled correlation coefficient estimate for PWv and SAF was 0.24 (95% CI: 0.15, 0.33), with moderate heterogeneity (I2: 72.7%; *p* = 0.000). For C-IMT and SAF, the pooled correlation coefficient estimate was 0.35 (95% CI: 0.27, 0.44), with moderate heterogeneity (I2 = 52.5%; *p* = 0.017). 

Meta-regression models of random effects for PWv showed significant results for age (*p* = 0.007) and HbA1c (*p* = 0.004). Age (*p* = 0.004) and SBP (*p* = 0.08) were significant moderators of the association between C-IMT and SAF (Appendix A).

### 3.5. Publication Bias

Evidence of publication bias was observed through funnel plot asymmetry and the Egger´s test for both PWV and C-IMT with SAF (*p* = 0.003 and 0.052, respectively).

## 4. Discussion

The relationship between early markers of vascular dysfunction, such as PWv or C-IMT and SAF, suggests that this novel method may be related to arterial stiffness and atherosclerosis. This systematic review and meta-analysis provide a synthesis of the evidence, showing a positive weak association between PWv and SAF (pooled *r* = 0.25; 95% CI: 0.18, 0.31), and between C-IMT and SAF (pooled *r* = 0.31; 95% CI: 0.25, 0.38).

Most studies analyzing the association between SAF and arterial stiffness reported that AGEs were associated with an increase in PWv in healthy individuals, patients with end-stage renal failure, hypertension, and type 2 diabetes mellitus [14,15,16,17,18,19]. The pathophysiological mechanisms responsible for increased arterial stiffness remain unclear. There are two main mechanisms through which arterial stiffness can be favored by the accumulation of AGEs [48]: (1) A complex interaction between functional and structural changes in the arterial wall, which lead to an overproduction of collagen and a decrease in the amount of elastin. These are both responsible for a reduction in arterial compliance properties, resulting in increased arterial stiffness [49]. (2) Receptors of AGEs (RAGEs) trigger different signaling pathways, resulting in the activation of nuclear transcription factors, and the secretion of proinflammatory cytokines and vascular adhesion molecules, which stimulate the progression of atherosclerosis [50].

Our results are in line with those of previous studies reporting a positive association between AGEs and C-IMT [22]. Several studies described the usefulness of AGE measurement using SAF as an indicator of changes in the thickness of the intima media of the carotid artery in the early stages of atherosclerosis [13]. AGEs are associated with vascular complications due to a change in the three-dimensional structure of proteins by crosslinking: (1) The accumulation of AGEs reduces vasodilation by decreasing nitric oxide levels and improves vasoconstriction by increasing endothelin-1 levels; additionally, this accumulation will cause the AGEs of the extracellular matrix to be modified accelerating the progression of the disease [51]. (2) The binding of AGEs with their RAGEs causes changes in the phenotype of different cells, such as endothelial cells, pericytes, or smooth muscle cells [52]. (3) The AGEs–RAGEs complex activates nuclear transcription factors and proinflammatory cytokine secretion, producing endothelial adhesion molecules, which positively influence the development of atherosclerosis [53].

AGEs may play an essential role in endothelial dysfunction and vascular inflammation, which may be a consequence of the above mentioned mechanisms. AGEs activate different proinflammatory routes triggering oxidative stress, inflammation, and apoptosis, resulting in arterial stiffness and atherosclerosis [48,53].

Traditional risk factors, such as age, glucose levels, or blood pressure, also contribute to arterial stiffness and atherosclerosis [54,55] and are related to AGE levels [6]. However, as reported in our results, although age, SBP, and HbA1c act are modifiers of the effect, there was a remaining positive association of PWv and C-IMT and SAF.

Our systematic review and meta-analysis have some limitations that should be stated. First, since the association reported by the included studies are cross-sectional in nature, a cause–effect relationship cannot be inferred. Therefore, it seems important that future follow up studies examine this cause–effect relationship between PWv and C-IMT with SAF. Second, most studies showed a moderate risk of heterogeneity; therefore, our results should be interpreted with caution. Third, there was evidence of publication bias by Egger’s test and unpublished results could modify the findings of this meta-analysis. Fourth, most studies provided neither information on the blinding of evaluators nor the justification of the necessary sample size, which may have led to bias. Last, participants with different health conditions (i.e., chronic kidney disease, hypertension, and diabetes) were included in the meta-analysis and this could have biased the results.

## 5. Conclusions

In summary, our results support that the increases in PWv and C-IMT, which are measures of arterial stiffness and atherosclerosis, respectively, are associated with the increase in SAF. Although the appropriate use of our results should be understood in each particular clinical context, our data suggest that clinicians may consider AGE levels measured by SAF when they assess the early stages of cardiovascular risk. Notwithstanding, our data highlight the need for more research to establish an optimal level of SAF in different populations to evaluate the appropriateness of including this biomarker as a routine assessment in clinical practice for cardiovascular risk among patients with different CVD risk levels. 

## Figures and Tables

**Figure 1 ijerph-17-06936-f001:**
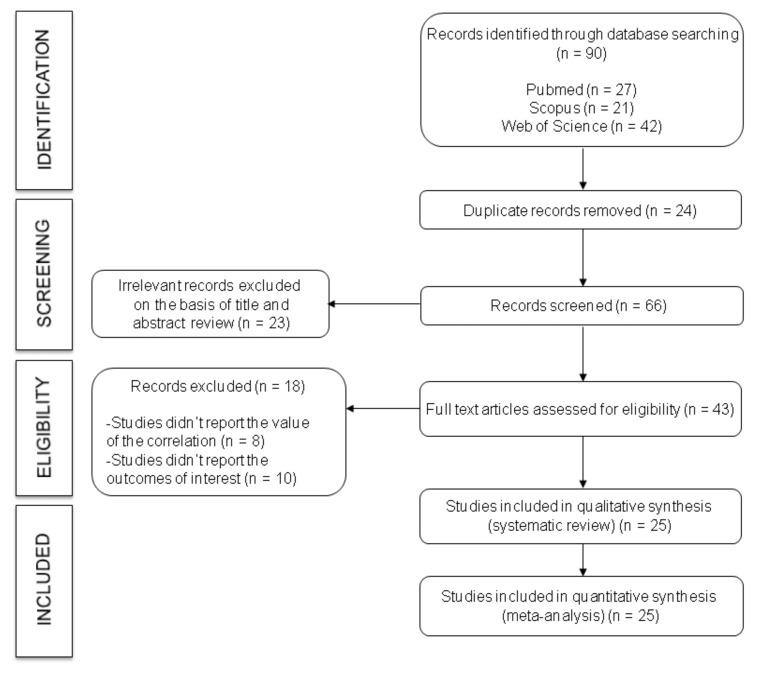
Flowchart: Search strategy.

**Figure 2 ijerph-17-06936-f002:**
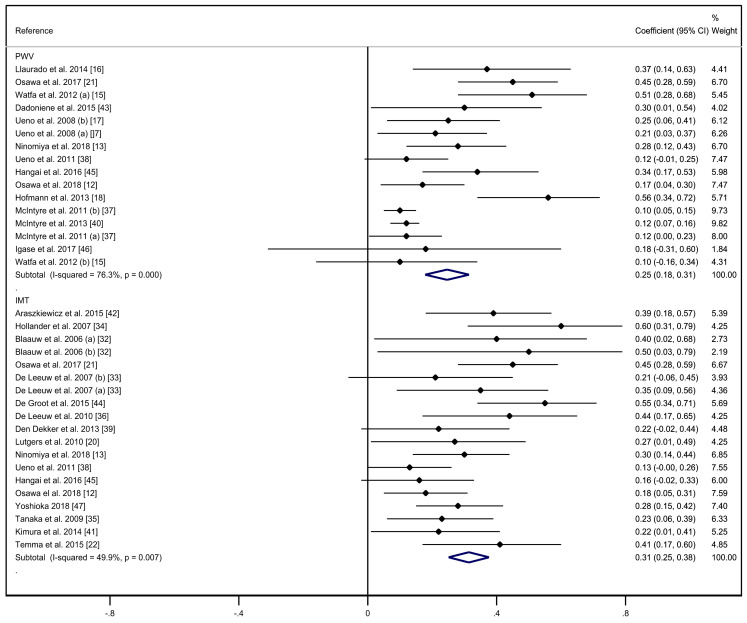
Forest plot including correlation between pulse wave velocity or carotid intima media thickness and skin autofluorescence.

**Figure 3 ijerph-17-06936-f003:**
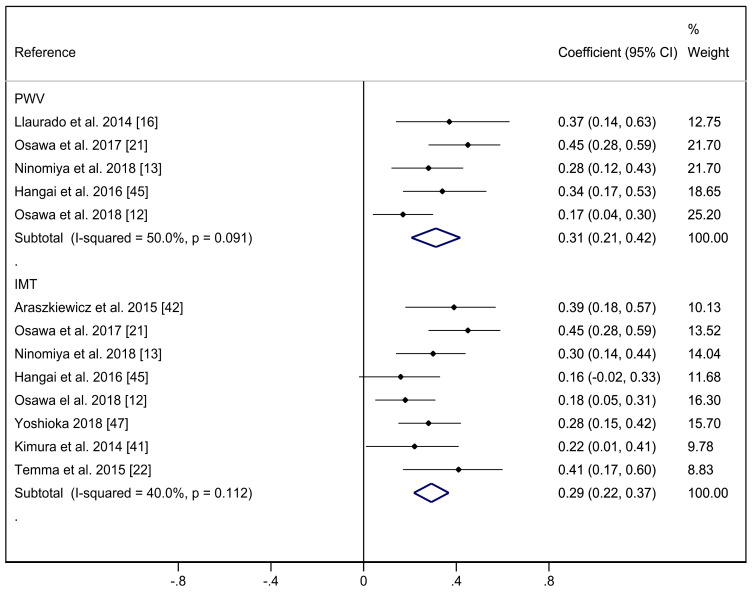
Forest plot including correlation between pulse wave velocity or carotid intima media thickness and skin autofluorescence in diabetic patients.

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
