# Peer review of "Are Advanced Glycation End Products in Skin Associated with Vascular Dysfunction Markers? A Meta-Analysis"

_ijerph, 2020, doi:10.3390/ijerph17186936_

Round 1
Reviewer 1 Report
In this manuscript, Alicia Saz-Lara and colleagues provided the systematic review and meta-analysis of the association between skin AGE and subclinical atherosclerosis marker, ie., PWv and C-IMT.
The measurement of skin autofluorescence is non-invasive, and associated with endothelial dysfunction and vascular aging. In the future, the measurement of skin autofluorescence may be used for early detection of atherosclerosis. However, there has been no systematic review and meta-analysis on this theme in the past. This work is well performance and the manuscript is carefully written. Thus, I think this manuscript deserves accept. Nevertheless, there are some minor issues that needed to be written.
Minor comments
- Line 17, 69, the authors described “March 30”. What year is March 30?
- Line 34, the authors described “accumulation of AGEs in tissue and vessels are measured by skin autofluorescence”. However, AGE reader can detect skin (tissue) AGE, not vessel AGE, I think. Then, accurate description is desired.
- References 14-17 in the introduction section show the relationship between skin autofluorescence and PWV. While, references 18 and 46-48 in the discussion section also show the relationship between skin fluorescence and PWV. Despite the same content, the references are different, so please unify.
- In ref 46, only serum carboxymethyl-lysine (CML) was evaluated as an AGE. Skin fluorescence was not evaluated. Therefore, reference 12 on healthy people should be replaced with other reference.
- Line 48-49, the authors mentioned the association between skin autofluorescence and PWV in patients with coronary heart disease, type 1 diabetes, or kidney disease. How about type 2 diabetes? The Maastricht Study (PMID: 27550921) showed the association between skin autofluorescence and PWVs was more pronounced in patients with type 2 diabetes.
- Line 52, the authors described “C-IMT is no longer included as a parameter of CVD risk-------“, and cited reference 21. However, reference 21 is not suitable because this is not international guideline. AHA/ACC guideline 2013 year recommend against the use of carotid IMT for individual risk prediction in clinical practice (J Am Coll Cardiol. 2014;63(25 Pt B):2935–2959).
Author Response
Reviewer 1
- Line 17, 69, the authors described “March 30”. What year is March 30?
Authors:
We apologize for the mistake. As suggested, we have modified the date.
In the Abstract section:
“A systematic search was performed using: MEDLINE (PubMed), SCOPUS and Web of Science, until 30 March 2020”.
In the Material and Methods section:
“A systematic search of studies was carried out through three databases: MEDLINE (via PubMed), Scopus and Web of Science, from their inception to 30 March 2020 (Table S1)”.
- Line 34, the authors described “accumulation of AGEs in tissue and vessels are measured by skin autofluorescence”. However, AGE reader can detect skin (tissue) AGE, not vessel AGE, I think. Then, accurate description is desired.
Authors:
The reviewer´s comment is correct. Thus, we have removed "vessels" from the sentence.
“Evidence suggests that the accumulation of AGEs in tissues, measured by skin autofluorescence (SAF) [4], increases with age and smoking [5], as well as in individuals with high levels of inflammation or diabetic conditions [6]”.
- References 14-17 in the introduction section show the relationship between skin autofluorescence and PWV. While, references 18 and 46-48 in the discussion section also show the relationship between skin fluorescence and PWV. Despite the same content, the references are different, so please unify.
Authors:
We really appreciate the reviewer’s thoughtful comment. As suggested, we have unified these references from the introduction and discussion sections.
References:
- Amar, J.; Ruidavets, J.B.; Chamontin, B.; Drouet, L.; Ferrières, J. Arterial stiffness and cardiovascular risk factors in a population-based study. J Hypertens 2001,19,381-7.
- Watfa, G.; Soulis, G.; Tartagni, E.; Kearney-Schwartz, A.; Borghi, C.; Salvi, P.; et al. Relationship between tissue glycation measured by autofluorescence and pulse wave velocity in young and elderly non-diabetic populations. Diabetes Metab 2012,38,413-9.
- Llauradó, G.; Ceperuelo-Mallafré, V.; Vilardell, C.; Simó, R.; Gil, P.; Cano, A.; et al. Advanced glycation end products are associated with arterial stiffness in type 1 diabetes. J Endocrinol 2014,221,405-13.
- Ueno, H.; Koyama, H.; Tanaka, S.; Fukumoto, S.; Shinohara, K.; Shoji, T.; et al. Skin autofluorescence, a marker for advanced glycation end product accumulation, is associated with arterial stiffness in patients with end-stage renal disease. Metabolism 2008,57,1452-7.
- Hofmann, B.; Adam, A.C.; Jacobs, K.; Riemer, M.; Erbs, C.; Bushnaq, H.; et al. Advanced glycation end product associated skin autofluorescence: A mirror of vascular function? Exp Gerontol 2013,48,38-44.
- Van Eupen, M. G. A.; Schram, M. T.; Van Sloten, T. T.; Scheijen, J.; Sep, S. J. S.; Van del Kallen, C. J.; et al. Skin Autofluorescence and Pentosidine Are Associated With Aortic Stiffening. The Maastricht Study. Hypertension 2016, 68,956–963.
- In ref 46, only serum carboxymethyl-lysine (CML) was evaluated as an AGE. Skin fluorescence was not evaluated. Therefore, reference 12 on healthy people should be replaced with other reference.
Authors:
Thank you for the reviewer’s comment. As suggested, we have replaced the mentioned reference by the following one:
Reference:
- Watfa, G.; Soulis, G.; Tartagni, E.; Kearney-Schwartz, A.; Borghi, C.; Salvi, P.; et al. Relationship between tissue glycation measured by autofluorescence and pulse wave velocity in young and elderly non-diabetic populations. Diabetes Metab 2012,38,413-9.
- Line 48-49, the authors mentioned the association between skin autofluorescence and PWV in patients with coronary heart disease, type 1 diabetes, or kidney disease. How about type 2 diabetes? The Maastricht Study (PMID: 27550921) showed the association between skin autofluorescence and PWVs was more pronounced in patients with type 2 diabetes.
Authors:
The reviewer´s comment seems judicious. As suggested, we have included the proposed reference.
“The accumulation of AGEs has been associated with an increase in pulse wave velocity (PWv), which is the gold standard for measuring arterial stiffness [13], both in healthy adults and individuals with different pathologies, such as patients with coronary heart disease, type 1 or 2 diabetes mellitus, and kidney disease [14-19]”.
Reference:
- Van Eupen, M. G. A.; Schram, M. T.; Van Sloten, T. T.; Scheijen, J.; Sep, S. J. S.; Van del Kallen, C. J.; et al. Skin Autofluorescence and Pentosidine Are Associated With Aortic Stiffening. The Maastricht Study. Hypertension 2016, 68,956–963.
- Line 52, the authors described “C-IMT is no longer included as a parameter of CVD risk-------“, and cited reference 21. However, reference 21 is not suitable because this is not international guideline. AHA/ACC guideline 2013 year recommend against the use of carotid IMT for individual risk prediction in clinical practice (J Am Coll Cardiol. 2014;63(25 Pt B):2935–2959).
Authors:
Thank you for the reviewer´s comment. As suggested, we have modified the reference.
Reference:
- Goff, D. C. ; Lloyd-Jones, D. M.; Bennett, G.; Coady, S.; D’Agostino, R. B.; Gibbons, R.; et al. 2013 ACC/AHA guideline on the assessment of cardiovascular risk: a report of the American College of Cardiology/American Heart Association Task Force on Practice Guidelines. J Am Coll Cardiol 2014,63(25 Pt B),2935–2959.
Reviewer 2 Report
Overall summary:
This manuscript provide a systematic review and meta-analysis to show the evidence regarding the association of arterial stiffness measured by pulse wave velocity and atherosclerosis measured by carotid intima media thickness with skin autofluorescence. In their analyses, the authors have showing a positive weak association of pulse wave velocity and carotid intima media thickness with skin autofluorescence. Overall, while this article is interesting and could potentially be of significance in cardiovascular diseases, there are significant points that the authors could clarify better. At this stage, this article would benefit from a major editing before publishing.
Major Comments:
- Appropriate control group (UNHS vs. HS) is not consistent in the selected studies, and it is a criterion that was not included in the study design. In fact, 9 out of 25 studies analyzed lack appropriate control group.
- This manuscript has intensively relayed on cross sectional studies, which are not as informative about the cause-and-effect relationships, when compared to the longitudinal studies (only three studies), especially that health conditions analyzed were not the same.
Minor Comments:
- The manuscript is poorly written and not easy to read. Throughout the manuscript there are grammar errors and misspelled words. For example the authors stated, “This systematic review and meta-analysis was reported according to the Meta-analysis of Observational Studies in Epidemiology statement (MOOSE) [23] and the Preferred Reporting Items for Systematic Reviews and Meta-analyses (PRISMA) [24], and performed following the Cochrane Collaboration Handbook recommendations [25].”
- Authors have explained in details the search strategy section, but have failed to clearly the year period of which the data were collected. Page2, line 69: Clarify years in which search have taken place “...from their inception to March 30.”
- References have not been used properly throughout the manuscript. For examples:
Page 3, line 132: two studies are missing “The search retrieved a total of 90 articles, of which, 25 studies [11,12,14-20,30-45] were selected and included in the systematic review according to the inclusion criteria.”
Page 11, line 218: missing reference “the binding of AGEs with their RAGEs causes changes in the phenotype of different cells, such as endothelial cells, pericytes or smooth 217 muscle cells.”
- Page 3, line 133: incorrect statement, “...Finally, 13 studies evaluating the association between PWv and SAF...” The numbers of PWv and SAF studies don’t not add-up, according to table 1.
- Table 1, second from the bottom: please correct typo “Japal”.
Author Response
Reviewer 2
Major Comments:
- Appropriate control group (UNHS vs. HS) is not consistent in the selected studies, and it is a criterion that was not included in the study design. In fact, 9 out of 25 studies analyzed lack appropriate control group
Authors:
The reviewer´s comment seems judicious. As suggested, to analyze only the studies that included a control group, we have performed a sensitivity analysis excluding the 9 studies without a control group. According to this, the following new sentences have been included in the manuscript:
In the Material and Methods section:
“Additionally, a sensitivity analysis was performed including only studies with a control group”.
In the Results section:
“Figure S1 shows the correlation coefficients between PWv and SAF, and C-IMT and SAF for the analyses that included only studies with a control group. The pooled correlation coefficient estimate for PWv and SAF was 0.24 (95%CI: 0.15, 0.33), with moderate heterogeneity (I2: 72.7%; p = 0.000). For C-IMT and SAF, the pooled correlation coefficient estimate was 0.35 (95%CI: 0.27, 0.44), with moderate heterogeneity (I2 = 52.5%; p = 0.017)”.
In the Supplemental material:
“Figure S1. Forest plot including the correlation between pulse wave velocity or carotid intima media thickness and skin autofluorescence in studies that included a control group”.
- This manuscript has intensively relayed on cross sectional studies, which are not as informative about the cause-and-effect relationships, when compared to the longitudinal studies (only three studies), especially that health conditions analyzed were not the same.
Authors:
Thank you for the reviewer’s thoughtful comment. As suggested, a cause-and-effect relationship can not be established. Furthermore, unfortunately, the three studies with longitudinal design included in this meta-analysis did not analyze prospectively the relationship between the C-IMT and the SAF, so it was only possible to analyze the correlation in the baseline measurement of both parameters. Thus, we have included the following sentence acknowledging this limitation in the discussion section.
“First, since the association reported by the included studies are cross-sectional in nature, a cause–effect relationship cannot be inferred. Therefore, it seems important that future follow up studies examine this cause–effect relationship between PWv and C-IMT with SAF”.
Minor Comments:
- The manuscript is poorly written and not easy to read. Throughout the manuscript there are grammar errors and misspelled words. For example the authors stated, “This systematic review and meta-analysis was reported according to the Meta-analysis of Observational Studies in Epidemiology statement (MOOSE) [23] and the Preferred Reporting Items for Systematic Reviews and Meta-analyses (PRISMA) [24], and performed following the Cochrane Collaboration Handbook recommendations [25].”
Authors:
Thank you for the reviewer’s comment. We would like to apologize for the grammar errors and misspelled words. We have sent the manuscript for English proofreading.
- Authors have explained in details the search strategy section, but have failed to clearly the year period of which the data were collected. Page2, line 69: Clarify years in which search have taken place “...from their inception to March 30.”
Authors:
Thank you for the reviewer’s comment. As suggested, we have modified the date.
“A systematic search of studies was carried out through three databases: MEDLINE (via PubMed), Scopus and Web of Science, from their inception to 30 March 2020 (Table S1)”.
- References have not been used properly throughout the manuscript. For examples:
- Page 3, line 132: two studies are missing “The search retrieved a total of 90 articles, of which, 25 studies [11,12,14-20,30-45] were selected and included in the systematic review according to the inclusion criteria”.
Authors:
Thank you for the reviewer’s comment. As suggested, we have included the two missing references.
“The search retrieved a total of 90 articles, of which, 25 studies [12,13,15-18,20-22,32-47] were selected and included in the systematic review according to the inclusion criteria”.
-
- Page 11, line 218: missing reference “the binding of AGEs with their RAGEs causes changes in the phenotype of different cells, such as endothelial cells, pericytes or smooth 217 muscle cells”.
Authors:
Thank you for the reviewer’s comment. As suggested, we have included the following reference:
- Ferland-McCollough, D.; Slater, S.;Richard, J.; Reni, C.; Mangialardi, G. Pericytes, an overlooked player in vascular pathobiology. Pharmacol Ther 2017,171,30–42.
- Page 3, line 133: incorrect statement, “...Finally, 13 studies evaluating the association between PWv and SAF...” The numbers of PWv and SAF studies don’t not add-up, according to table 1.
Authors:
Thank you for the reviewer’s comment. As suggested, we have modified this section according to table 1.
“Finally, 12 studies evaluating the association between PWv and SAF, and 17 studies evaluating the association between C-IMT and SAF were identified and considered in the meta-analysis”.
- Table 1, second from the bottom: please correct typo “Japal”.
Authors:
Done. Thank you.